# Survival Outcome of Empirical Antifungal Therapy and the Value of Early Initiation: A Review of the Last Decade

**DOI:** 10.3390/jof8111146

**Published:** 2022-10-29

**Authors:** Souha S. Kanj, Ali S. Omrani, Hail M. Al-Abdely, Ahmad Subhi, Riad El Fakih, Ibraheem Abosoudah, Hazar Kanj, George Dimopoulos

**Affiliations:** 1Division of Infectious Diseases, Department of Internal Medicine, American University of Beirut Medical Center, Riad El Solh, Beirut P.O. Box 11-0236, Lebanon; 2Division of Infectious Diseases, Department of Medicine, Hamad Medical Corporation, Doha P.O. Box 3050, Qatar; 3College of Medicine, Qatar University, Doha P.O. Box 2713, Qatar; 4Division of Infectious Diseases, Department of Medicine, King Faisal Specialist Hospital and Research Centre, Riyadh 11211, Saudi Arabia; 5Division of Infectious Disease, Al-Qassimi Hospital, Emirates Health Services, Sharjah 61313, United Arab Emirates; 6Department of Hematology, Stem Cell Transplant & Cellular Therapy, King Faisal Specialist Hospital and Research Centre, Riyadh 11211, Saudi Arabia; 7Department of Oncology, King Faisal Specialist Hospital and Research Center, MBC J-64, Jeddah 21499, Saudi Arabia; 8Faculty of Medicine, American University of Beirut Medical Center, Beirut P.O. Box 11-0236, Lebanon; 9Department of Critical Care, “EVGENIDIO” Hospital, National and Kapodistrian University of Athens (NKUA), 10679 Athens, Greece

**Keywords:** intensive care, empirical therapy, antifungal agents, invasive candidiasis, invasive aspergillosis

## Abstract

Aim: This rapid systematic review aimed to collect the evidence published over the last decade on the effect of empirical antifungal therapy and its early initiation on survival rates. Methods: A systematic search was conducted in PubMed, Cochrane, Medline, Scopus, and Embase, in addition to a hand search and experts’ suggestions. Results: Fourteen cohort studies and two randomized clinical trials reporting the survival outcome of empirical antifungal therapy were included in this review. Two studies reported the association between early empirical antifungal therapy (EAFT) and survival rates in a hematological cancer setting, and fourteen studies reported the outcome in patients in intensive care units (ICU). Six studies reported that appropriate EAFT decreases hospital mortality significantly; ten studies could not demonstrate a statistically significant association with mortality rates. Discussion: The inconsistency of the results in the literature can be attributed to the studies’ small sample size and their heterogeneity. Many patients who may potentially benefit from such strategies were excluded from these studies. Conclusion: While EAFT is practiced in many settings, current evidence is conflicting, and high-quality studies are needed to demonstrate the true value of this approach. Meanwhile, insights from experts in the field can help guide clinicians to initiate EAFT when indicated.

## 1. Introduction

Four decades ago, Pizzo et al. first suggested empirical antifungal therapy (EAFT) for cancer patients with granulocytopenia to control occult invasive fungal infections and prevent new ones [1]. The study found that adding amphotericin B to the standard empirical antibacterial agents significantly decreased morbidity and mortality [1]. While the recent progress in diagnostic tools has led some to question the future role of EAFT, recent evidence has shown conflicting results [2]. Moreover, invasive fungal infections (IFIs) continue to be associated with unacceptably high morbidity and mortality rates worldwide [3]. Hence, many clinicians continue to give EAFT in various settings. Nevertheless, the emergence of resistance among *Candida* and *Aspergillus* spp. over the years, the cost of the drugs, and the potential side effects of antifungal (AF) agents should be in line with the principles of antimicrobial stewardship.

### 1.1. Epidemiology

A review of the epidemiology of fungal diseases published five years ago revealed an increasing incidence and mortality rate comparable to that of tuberculosis [3,4]. Invasive candidiasis (IC) is the most prevalent IFI worldwide, followed by invasive aspergillosis (IA) with about 700,000 cases and 250,000 confirmed annual cases, respectively [3,4]. More recently, IA has been increasingly reported in patients with influenza or SARS-CoV-2 infection and *Candida auris* has newly emerged in many countries worldwide, especially following the COVID-19 pandemic [5]. Despite all the therapeutic advances, the mortality rate among patients with candidemia remains around 25%, and 30 to 80% for patients with IA [6,7]. Globally, an estimated 1.6 million deaths occur annually from IFIs, suggesting that perhaps delays in therapy might play a contributing role and raise the question of whether early empirical therapy may improve patients’ poor outcomes [3].

### 1.2. Diagnostic Methods

Historically, the definitive diagnosis for IFI relied mainly on histopathological evidence or cultures of sterile sites [8]. Culture (the gold standard for confirming IFI) and microscopy-based diagnoses of IFI have a slow turn-around time, low sensitivity for candidiasis, and specificity for aspergillosis/invasive mold infections [9]. Patients at risk for IFIs are usually already at an increased risk of rapid clinical deterioration and death. Delaying initiation of AF agents until the culture results are available may significantly worsen these patients’ outcomes. Therefore, strategies to improve timely diagnostic confirmation of IFIs should be based on rapid screening tests, including biomarkers and molecular tests, to enable earlier appropriate AF therapy and improve clinical outcomes [7,10]. However, there remain limitations in the sensitivity and specificity of the current diagnostic tests which are not available in many countries, including in centers caring for high-risk patients [11,12,13]. This precludes the timely detection of IFIs in many settings [14]. As a result, many IFIs go unrecognized, and AFT is often not administered promptly [15].

The most used biomarker-based diagnostics include β-d-glucan and galactomannan assays. A systematic review published in 2016 found these biomarkers to have variable sensitivity, specificity, and positive predictive values [11]. The negative predictive value (NPV) for galactomannan assay was found to exceed 70% and reach up to 100% [16]. In intensive care unit (ICU) patients at increased risk for candidiasis, the NPV of both β-d-glucan and polymerase chain reaction (PCR) is high (≥95%) and increases up to >99% in cases of fever or sepsis [14]. Similarly, for aspergillosis, the NPV for PCR in immunocompromised neutropenic patients is high [8]. In settings where these tests are available, they can be of value to guide EAFT. However, their usefulness is still debatable in pediatric patients [17]. Overall, when AFT is indicated, initiating therapy can be guided by clinical suspicion based on the severity of the illness and underlying risk factors.

### 1.3. Definitions

So far, there is no consensus regarding the terminology of the various systemic antifungal treatment strategies with different guidelines for different settings [18]. For the purpose of this review, AF therapy will be classified into three major categories regardless of the setting and patient population (Figure 1). The first category is prophylaxis which is indicated for patients assumed to be free of but at risk for IFIs (no suggestive symptoms of IFI) [19]. The second is empirical for patients suspected to have IFIs regardless of the spectrum of the agent used or whether there are one or more target pathogens considered in the differential diagnosis [20]. In septic ICU patients on antibacterial agents and in whom candidiasis is suspected, the likelihood of having IFIs can range from 20 to 90%, depending on the patient’s profile [21]. This uncertainty and wide range of probability make the decision to initiate treatment challenging and highlight the importance of timely initiation of EAFT [9]. The third is diagnostic-based therapy which encompasses the diagnostic-driven (in patients having results from biomarker and/or molecular assays suggestive of a higher likelihood of an IFI) and the definitive (in patients with a confirmed IFI as a result of culture, histopathology or microscopic examination) [22,23,24]. Hence, in this review, data from patients with radiological evidence were included, whereas data from patients with any other diagnostic evidence were not considered eligible.

### 1.4. Purpose

Due to the poor outcomes associated with IFIs, most clinicians prefer to start early EAFT rather than waiting for the diagnosis to be confirmed. Studies have shown that delaying the treatment of candidemia is associated with increased mortality [25]. In addition, a retrospective study showed that inappropriate voriconazole therapy was associated with increased mortality in IA due to resistant *Aspergillus* spp. [26]. However, there remain uncertainties as to whether early EAFT is the best approach due to the limitations of the current diagnostics and the complexity of creating a high-quality design study with a large homogenous population, which impair the ability to produce statistically significant results. This rapid systematic review aims at analyzing the evidence from the last decade on the outcome of EAFT and the value of early initiation to support the practice and reveal the gaps in the literature.

## 2. Materials and Methods

The present systematic rapid review conforms to the Preferred Reporting Items for Systematic Reviews and Meta-Analyses (PRISMA) 2020 recommendations [27].

### 2.1. Search Databases and Terms

Scopus, Medline (Ovid), PubMed, Embase, and Cochrane Library were searched for randomized controlled trials (RCT) and cohort studies reporting the survival rate among patients administered EAFT. In addition, a hand search was carried out, and experts in the field were contacted for additional article suggestions. Only articles published during the past decade (January 2012 to January 2022) were included. Search terms used were Candidemia, Sepsis, Fungemia, Candidiasis, Antifungal Agents, Azoles, Amphotericin, Echinocandins, and Empirical. Both keywords and MESH terms (index words) were used when available. The search terms and strategies on each database can be found in Appendix A and Appendix B, respectively. Limits did not restrict by language. However, all of the articles that passed the screening were in English.

### 2.2. Selection Criteria and Screening

Two major inclusion criteria were used during the screening. The first is the original peer-reviewed RCT or cohort. The second is reporting explicitly or implicitly the survival rate of patients on EAFT or reporting on the association between early EAFT administration and the survival rate. Review articles, case reports, letters, and editorials were excluded. Studies that included patients with confirmed diagnosis before treatment initiation or did not report the survival rate of patients who received EAFT neither directly nor indirectly were excluded.

Titles and abstracts screening was carried out by one reviewer using the JBI SUMARI platform. To minimize selection bias at this stage, every RCT or cohort study reporting the outcomes of empirical therapy was included for full-text screening. Two reviewers screened the full text for eligibility. Studies in which the diagnosis was known before administering the antifungal agent were excluded even if the treatment was labeled as empirical. Studies investigating the outcomes of AFT given to symptomatic patients before having diagnostic evidence of IFI were included, irrespective of whether the therapeutic strategy was identified as empirical or not.

### 2.3. Critical Appraisal and Data Extraction

Included articles were assessed for risk of bias by two reviewers using the Joanna Briggs Institute (JBI) checklists for RCTs and cohort studies (see Appendix C for the checklist questions). Studies with a high risk of bias were not excluded, but their limitations were discussed.

The primary endpoints of the present review are the survival rate in patients who were given EAFT and whether EAFT was independently correlated with the survival rate. In studies in which this outcome was not explicitly stated, a calculation from the given data was made. In studies comparing the outcome of diagnostic-based AF treatment to that of EAFT, both outcomes were extracted and reported in this review. A meta-analysis was not performed because of the heterogeneity of the studies.

## 3. Results

### 3.1. Records Screening

Out of 2733 individual records, 49 articles were included at the title and abstract screening, and 46 were retrieved for full-text screening (Figure 2). Out of these, only 14 met the inclusion criteria. Two additional articles were found eligible from the ones identified by hand search. Experts in the field identified four additional articles, all of which were retrieved, and 1 article was added to the final selection. This review reports EAFT survival outcomes from 16 original studies [28,29,30,31,32,33,34,35,36,37,38,39,40,41,42,43].

### 3.2. Risk of Bias Assessment

The appraisal results for the RCTs and Cohort studies are reported in Table 1 and Table 2, respectively. Overall, the quality of the studies was considered acceptable, and no study was excluded at this point. The effect of the follow-up period on the survival rate was found to be significant. In the study carried out by Montravers et al., the survival rate was reported at different points in time, revealing that while the survival rate upon discharge from ICU was 71%, it decreased to 59% when the endpoint was defined as hospital discharge [35]. Therefore, the follow-up duration is reported in Table 2 for better comparison.

### 3.3. Data Extraction

Among the 16 original studies reported in this review, 2 are RCTs, 13 are retrospective cohort studies, and 1 is a prospective cohort study (Table 3 and Table 4).

In critically ill patients admitted to the ICU, twelve studies from seven countries (France, USA, Japan, Spain, Tunisia, Italy, and Portugal) reported the survival rate for patients administered EAFT. Nine studies did not show a statistically significant association between the administration of EAFT and their respective survival rates, while three retrospective cohorts reported a significant association between survival rate and early EAFT administration. A study by Kollef et al. found that delayed AFT, defined as no antifungal therapy within 24 h of the onset of septic shock, was independently associated with greater odds of in-hospital mortality when compared to earlier EAFT administration and appropriate source control (adjusted odds ratio 33.75; 95% CI: 9.65–118.04, *p* = 0.005) [33]. Similarly, Tedeschi et al. found that adequate EAFT administered within 72 h from blood drawing is independently associated with a reduced risk of in-hospital mortality (hazard ratio 0.42; 95% CI 0.25–0.69, *p* = 0.001). Finally, a study by Greenberg et al. on infants found that EAFT was associated with an increased survival rate without neurodevelopmental impairment compared to those not administered EAFT [31].

The two randomized, double-blind, placebo-controlled trials did not demonstrate improved survival with empirical echinocandin therapy in critically ill patients at increased risk of IFI. The study by Timsit et al. included 260 non-neutropenic ICU-acquired sepsis and multiple risk factors and compared the effect of micafungin empirical treatment to placebo [42]. Although the empirical treatment significantly decreased the rate of new fungal infections, it did not increase the survival rate at 28 days. The other RCT studied the effect on mortality as a secondary endpoint using caspofungin for 222 ICU patients.

In the hematological malignancies (HM) setting, only one eligible retrospective cohort study reported the outcomes of patients who received EAFT. A single-center study from the USA that enrolled 146 cancer patients with *Candida glabrata* candidemia, among which 99 (68%) patients had solid tumors and 47 (32%) had hematological malignancies. Early appropriate AFT, defined as receipt of an in vitro-active antifungal agent within 48 hours of blood culture collection, was independently associated with decreased 28-day all-cause mortality (adjusted HR 0.469, 95% CI 0.237–0.930, *p* = 0.03), and all-cause in-hospital mortality (adjusted OR 0.31; 95% CI 0.13–0.772, *p* = 0.011).

Overall, the survival rate in patients who were administered EAFT showed large variability ranging from 50 to 90%. Three studies compared mortality rates in patients administered EAFT to that of patients administered diagnostic-driven AFT. Among these, two studies from Singapore and China found that EAFT is an independent protective factor that decreases hospital mortality; and one study from Pakistan did not show a statistically significant association between EAFT administration and mortality rate compared to diagnostic-driven therapy.

## 4. Discussion

The present systematic review reported the results of 16 articles investigating the effect of EAFT on the survival of patients with IFIs. Ten studies did not demonstrate a statistically significant association between the administration of EAFT and survival rates, while the other six reported that early EAFT was superior to diagnostic-based treatment. Despite the inconsistent conclusions of the included studies, close inspection of the results reveals a common pattern. In contrast, 8 out of 10 studies found that the administration of EAFT did not increase the survival rate compared to no administration. The two studies that found statistically significant correlations did not include all patients on EAFT but those who were administered EAFT promptly (Table 3).

Whereas the lack of strong evidence on the value of EAFT administration can be explained by the lack of high-quality study designs with a large and homogenous patient population that has enough power to demonstrate significance, the likelihood of finding a statistically significant correlation increases significantly when discussing the value of early initiation as opposed to EAFT administration at any time. This calls for the conduction of additional studies that can confirm that the protective effect of EAFT against mortality rate is contingent upon the timely administration in order to guide the practice accordingly. It is important to note that an additional challenge to generate new data on EAFT is that empirical antifungal therapy is the “standard of care” for patients with granulocytopenia and fever, thus explaining the low number of studies in our review.

Previous systematic reviews investigating the effect of EAFT on patient outcomes also revealed inconsistent results among the studies [18]. The fact that EAFT is usually administered earlier to sicker patients compared to culture-directed therapy undermines the positive effect of early initiation compared to late therapy.

The inconclusiveness of clinical evidence on the value of early initiation of EAFT is due to multiple factors. Despite the valiant efforts, RCTs of empiric AFT have been limited by difficult enrollment, limited positive predictive values of the implemented tools, heterogeneous patient populations, and the possible enrollment of individuals with undetected baseline IFIs [44]. Therefore, the lack of evidence of survival benefits associated with EAFT should not necessarily be interpreted as evidence of lack of benefit. Notably, given the risk of adverse outcomes with delayed AFT in patients with suspected IFIs, current guidelines recommend early empirical antifungal therapy in high-risk patients in hematologic malignancies/HSCT settings as well as in some critically ill patients at high risk for severe disease and complications [45,46,47,48,49,50,51,52]. The heterogeneity of ICU patients and a large number of confounding factors make evidence-based recommendations for EAFT quite challenging.

The truth is that multiple trials, such as the EMPIRICUS trial, do not include patients who could have benefited from early EAFT administration, which underestimates the latter’s value and preclude drawing strong supporting evidence. This fact highlights the importance of using precision medicine when tackling IFIs empirically [53].

Considering the predominance of EAFT in practice, the literature still lacks strong evidence to support the value of such practice. Therefore, better study designs are needed to support, guide, and standardize the practice [54].

## 5. Strengths and Limitations

In this rapid systematic review, we conducted a comprehensive 10-year literature search, including five databases, a hand search, and experts’ suggestions which minimizes selection bias. The heterogeneity of the studies in terms of design, comparators, and outcome reported did not allow for a statistical synthesis of the results. All studies that report the outcome of interest were included regardless of the appraisal results and the conflict of interest. Finally, only outcomes of interest, namely, the survival rate in patients treated with EAFT, and the association of early initiation with hospital mortality, were extracted from the included studies because other outcomes are beyond the scope of this rapid review.

## 6. Conclusions

The current practice depends predominantly on EAFT to decrease morbidity and mortality in immunocompromised and critically ill patients. Several studies found that early initiation of EAFT, when indicated, improves patients’ outcome. However, the literature is still lacking strong evidence on the value of this practice. Studies of homogenous patient populations and which use new surrogate endpoints and accurate diagnostic methods are needed to support the current practice and facilitate early initiation of EAFT when indicated. Until further evidence is available, the medical community might benefit largely from global and regional experts’ insights on the indications of early EAFT, the barriers to early initiation and the criteria for drug selection to guide the practice.

## Figures and Tables

**Figure 1 jof-08-01146-f001:**
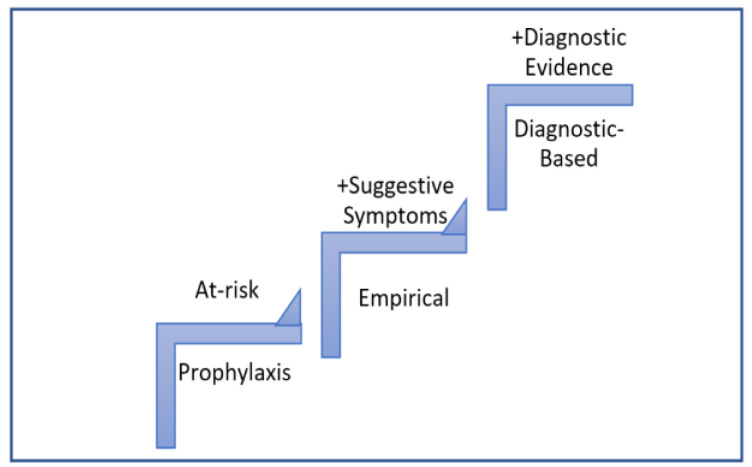
Definitions adopted in this review for the classification of antifungal therapy based on the certainty of the diagnosis.

**Figure 2 jof-08-01146-f002:**
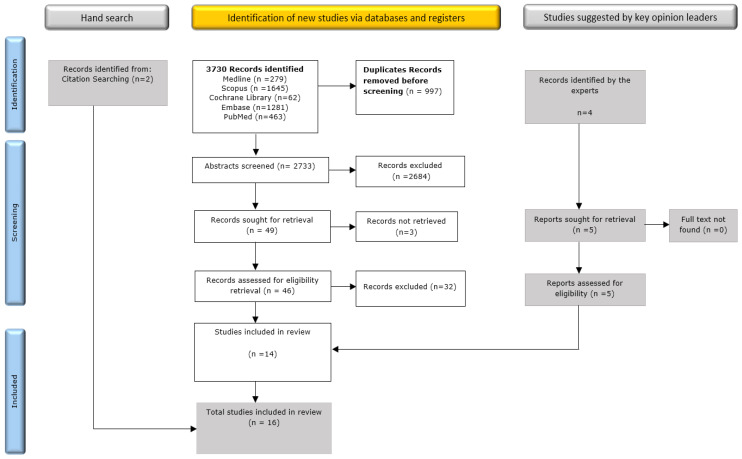
PRISMA Flow diagram of the search and screening results.

**Table 1 jof-08-01146-t001:** Randomized controlled trials critical appraisal results.

Citation	Q1 *	Q2	Q3	Q4	Q5	Q6	Q7	Q8	Q9	Q10	Q11	Q12	Q13
Ostrosky-Zeichner, et al. 2014 [37]	Y	Y	Y	Y	Y	Y	Y	Y	Y	Y	Y	Y	Y
Timsit, et al. 2016 [42]	Y	Y	Y	Y	Y	Y	Y	Y	Y	Y	Y	Y	Y

Y yes; U Unclear; N No; N/A Not applicable. * Q1–Q13 correspond to JBI critical appraisal tool for randomized controlled trials (Appendix C).

**Table 2 jof-08-01146-t002:** Cohort studies critical appraisal results.

Citation	Q1 *	Q2	Q3	Q4	Q5	Q6	Q7	F/U ^1^	Q9	Q10	Q11
Bailly, et al. 2015 [28]	N	N/A	**Y**	**Y**	**Y**	**Y**	**Y**	30	Y	Y	**Y**
Kato, et al. 2019 [32]	U	N/A	**Y**	**Y**	**Y**	**Y**	**Y**	30	Y	Y	**Y**
Kollef, et al. 2012 [33]	U	N/A	**Y**	**Y**	**Y**	**Y**	**Y**	TD/D	Y	Y	**Y**
Ohki, et al. 2020 [36]	Y	N/A	**Y**	**Y**	**Y**	**Y**	**Y**	30	Y	Y	**Y**
Poves-Alvarez, et al. 2019 [39]	N	N/A	**Y**	**Y**	**Y**	**Y**	**Y**	30	Y	Y	**Y**
Trifi, et al. 2019 [43]	Y	N/A	**Y**	**Y**	**Y**	**Y**	**Y**	28	Y	Y	**Y**
Tedeschi, et al. 2016 [41]	N	N/A	**Y**	**Y**	**Y**	**Y**	**Y**	TD/D	U	Y	**Y**
Greenberg, et al. 2012 [31]	Y	N/A	**Y**	**Y**	**Y**	**Y**	**Y**	TD/D	U	U	**Y**
Pinto-Magalhaes, et al. 2019 [38]	U	N/A	**Y**	**Y**	**Y**	**Y**	**Y**	30	Y	Y	**Y**
Montravers, et al. 2017 [35]	N	N/A	**Y**	**Y**	**Y**	**Y**	**Y**	28	Y	N	**Y**
Farmakiotis, et al. 2015 [30]	N	N/A	**Y**	**Y**	**Y**	**Y**	**Y**	28	Y	Y	**Y**
Lee, et al. 2014 [34]	U	N/A	**Y**	**Y**	**Y**	**Y**	**Y**	30	Y	Y	**Y**
Cui, et al. 2017 [29]	Y	N/A	**Y**	**Y**	**Y**	**Y**	**Y**	TD/D	U	U	**Y**
Raza, et al. 2016 [40]	U	N/A	**Y**	**Y**	**Y**	**Y**	**Y**	TD/D	N/A	N/A	**Y**

Abbreviations: Y yes; U Unclear; N No; N/A Not applicable; TD/D Till discharge from hospital /death. ^1^ Follow-up time in days. * Q1–Q11 correspond to JBI critical appraisal tool for cohort studies (Appendix C).

**Table 3 jof-08-01146-t003:** Characteristics of the Studies reporting the survival rate of patients administered EAFT and their main results.

Study	Country Study Design	Participant Characteristics	Survival Rate (n ^1^)	Description of Main Results
Bailly et al., 2015 [28]	France, Retrospective cohort	Non-neutropenic, nontransplant recipients, ICU, intubated for ≥5 days	70% (n = 100)	EAFT did not significantly reduce mortality in critically ill and mechanically ventilated patients
Ostrosky-Zeichner et al., 2014 [37]	USA, double-blind RCT	Adults in ICU ≥ 3 days, ventilated, received antibiotics, had a central line, and 1 additional risk factor ^2^	83% (n = 102)	Difference in mortality rate between caspofungin and placebo (20.5% vs. 15.7%) was not statistically significant (*p* = 0.39)
Timsit et al., 2016 [42]	France, double-blind RCT (EMPIRICUS)	Non-neutropenic, non-transplanted, critically ill patients with ICU-acquired sepsis, multiple Candida colonization, multiple organ failure, exposed to broad-spectrum AB agents	70.3% (n = 128)	The use of empirical micafungin in ICU settings did not decrease the mortality rate compared to placebo
Kato et al., 2019 [32]	Japan, retrospective cohort	Adults with blood culture yielded at least one Candida species	73.5% (n = 260)	Empiric treatment with fluconazole is significantly associated with 30-day mortality aOR= 0.32 95%CI 0.12–0.88 *p* = 0.026
Ohki et al., 2020 [36]	Japan, retrospective cohort	Adults in ICU with candidemia and central venous catheter in situ at the time of onset	(n = 62)	EAFT conferred no significant clinical benefit on survival
Poves-Alvarez et al., 2019 [39]	Spain, retrospective study	Non-neutropenic adults diagnosed with candidemia.	50% (n = 90)	Receiving EAFT did not result in a significant difference in the 30-day survival rate compared to no treatment.
Trifi et al., 2019 [43]	Tunisia, retrospective cohort	ICU non-neutropenic septic critically ill patients without proven fungal infection	64% (n = 45)	No significant beneficial impact of EAFT on 28- day survival as compared to non-EAFT (64 vs. 60%)
Tedeschi et al., 2016 [41]	Italy, retrospective cohort	Patients with candidemia in Internal Medicine Wards	64.46% (n = 166)	Adequate and timely ^3^ EAFT is an independent protective factor against in-hospital mortality (HR 0.42, 95%CI 0.25–0.69, *p* = 0.001)
Greenberg, 2012 [31]	USA, cohort study	Infants with birth weight >1 Kg and ≥1 positive culture for *Candida*	50% (n = 38)	EAFT was associated with increased survival without NDI
Montravers et al., 2017 [35]	France, prospective cohort	Patients in ICU treated for Candida peritonitis	69% (n = 204)	Survival at 28 days was not associated with empirical treatment

Abbreviations: AOR adjusted odds ratio; HR hazard ratio; NDI Neurodevelopmental impairment; EAFT Early empiric antifungal therapy. AB antibacterial. RCT randomized control trial, USA, United States of America. ^1^ n = Total number of patients administered EAFT among those included in the study. ^2^ Parenteral nutrition, dialysis, surgery, pancreatitis, systemic steroids, or other immunosuppressants. ^3^ Timely: within 72 h from the blood draw.

**Table 4 jof-08-01146-t004:** Studies comparing the outcome of empirical and diagnostic-based treatment, or early versus late EAFT administration.

Study	Country Study Design	Participants Characteristics	Survival Rate (%) ^1^		Description of Main Results
			Early EAFT	Late	
Lee et al., 2014 [34]	Singapore, prospective cohort	Adults in SICU s/p surgery ^2^	80% N = 30	50% N = 18	EAFT group were 4 times less likely to experience 30-day all-cause mortality than culture-directed (OR: 0.25, 95% CI: 0.069 to 0.905; *p* = 0.03)
Cui et al., 2017 [29]	China, retrospective cohort	Adult patients with proven ICI	69.7% N = 142	60.3% N = 73	EAFT was an independent predictor for DECREASING hospital mortality (OR 0.327, CI [0.160–0.667], *p* = 0.002
Raza et al., 2016 [40]	Pakistan, retrospective cohort	cancer patients with positive blood culture for Candida	N = 63	N = 165	Receipt of antifungal agents on an empirical basis was not significantly associated with mortality. AOR = 0.44 CI 0.18–1.12
Farmakiotis et al., 2015 [30]	USA, retrospective cohort	Cancer patients with candidemia *Candida glabrata*	N = 48	N = 98	Early appropriate AFT was independently associated with decreased 28-day mortality and in-hospital mortality (adjusted odds ratio 0.31, *p* 0.011). HR 0.374, 95% CI 0.197–0.709 (*p* = 0.003) ^3^. HR 0.357, 95% CI 0.178–0.718 (*p* = 0.004) ^4^
Kollef et al., 2012 [33]	USA, retrospective cohort	Hospitalized patients with septic shock and positive blood culture for Candida species	47.2% N = 142	2.4% N = 82	Delayed AFT is independently associated with a greater risk of hospital mortality ^5^. AOR 33.75; CI: 9.65–118.04 *p*= 0.005
Kato et al., 2019 [32]	Japan, retrospective cohort	Adults with blood culture yielded at least one Candida species	66%	61% N = 164	Earlier administration of appropriate AF therapy (within 48 h) might improve survival in ICU patients with candidemia, but the difference was not statistically significant
Pinto Magalhães 2019 [38]	Portugal, retrospective cohort	Adults with ≥1 positive culture for *Candida* species	72%	61.9% (Without AFT)	30 days-mortality rate is not significantly associated with the time between blood culture collection and the start of antifungal therapy

Abbreviations: ICIs Invasive Candida infections; AOR adjusted odds ratio. CI confidence interval. Note: ^1^ N = TOTAL sample size in each arm. ^2^ Surgery for gastrointestinal perforation, bowel obstruction or ischemia, malignancy, and anastomotic Leakages; SIRS despite AB. ^3^ After the removal of ID consultation from the model. ^4^ After excluding patients who died within 48 h after blood culture collection. ^5^ Delayed antifungal treatment (received no antifungal therapy within 24 h of the onset of shock).

## Data Availability

Not applicable.

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
