# Peer review of "Survival Outcome of Empirical Antifungal Therapy and the Value of Early Initiation: A Review of the Last Decade"

_jof, 2022, doi:10.3390/jof8111146_

Round 1

Reviewer 1 Report

I think this is an interesting article, but as authors mention, results on early empirical antifungal treatment are not conclusive.

Most of the papers reviewed in this case are devoted to invasive Candida infections. As it is stated, these infections are difficult to diagnose only by biomarkers.

On the other hand, other IFI sometimes can be diagnosed using direct examination of different samples (what is not said in the article), besides biomarkers and that allows to initiate an adequate early treatment.  

Tables 1 and 2. I consider authors should add that Q1 to Q10 correspond to JBI critical appraisal tool, there is no explanation for the meaning of the columns without that.

Author Response

The authors of the manuscript thank the reviewer for taking the time to inspect the manuscript and for the constructive and insightful comments, addressed as follow:

Point 1: Most of the papers reviewed in this case are devoted to invasive Candida infections. As it is stated, these infections are difficult to diagnose only by biomarkers. On the other hand, other IFI sometimes can be diagnosed using direct examination of different samples (what is not said in the article), besides biomarkers and that allows to initiate an adequate early treatment.  

Response 1: The direct microscopic examination have been added to the manuscript in multiple paragraphs with an explanation that similar to culutre it is limited by its low sensitivity.

Point 2: Tables 1 and 2. I consider authors should add that Q1 to Q10 correspond to JBI critical appraisal tool, there is no explanation for the meaning of the columns without that.

Response 2: Thank you very much for this perceptive comment, we have added the note to tables 1 and 2 as recommended.

Best regards.

Reviewer 2 Report

Kanj et al paper describes the impact of the antifungal early initiation vs the survival outcome by the literature review of the last decade.

It is an interesting approach in the evaluation of a very common clinical practice. The Authors' criticism is represents an useful point-of-view, suggesting the need of stronger evidences and clinical debates.

However, the Introduction has to be largely shortened, to better focus the literature exam.

Author Response

The authors of the manuscript thank the reviewer for taking the time to inspect the manuscript and for the constructive and insightful comments, addressed as follow:

Point 1: “The Introduction has to be largely shortened, to better focus the literature exam.”

Response 1: Thank you for this perceptive feedback. We have removed all redundancies and unnecessary information from the introduction. We did not remove more information as to keep the paper clear for general practitioners and doctors who might need this information to better understand the significance of the study and the application to the practice. Please let us know if you would like us to shorten the introduction further.

Best regards!